# Stress-Buffering Effects of Social Support on Tourism Employees during the COVID-19 Pandemic: A Moderated Mediation Model

**DOI:** 10.3390/ijerph20032342

**Published:** 2023-01-28

**Authors:** Huiyue Liu, Qiancai Tan, Huiping Mai

**Affiliations:** Department of Tourism Management, Shenzhen Tourism College, Jinan University, Shenzhen 518053, China

**Keywords:** social support, employment stress, tourism employees, psychological resilience, positive coping styles

## Abstract

Since the beginning of 2020, China’s tourism industry has been severely impacted by the COVID-19 pandemic, and domestic tourism revenues have plummeted. Tourism employees have faced reduced working hours, job instability, shut down, and unemployment. In the context of the normalization of epidemic prevention, the tourism industry is recovering slowly and uncertainly, and many tourism employees face increasing employment stress. To investigate the relationship between social support and employment stress among tourism employees, 308 tourism employees were surveyed, and the mediating role of positive coping styles and the moderating role of psychological resilience were examined using structural equation modeling. The results revealed three key findings: social support significantly and negatively influenced the employment stress of tourism employees; positive coping styles significantly mediated the relationship between social support and employment stress among tourism employees; and psychological resilience moderated the relationship between social support and employment stress among tourism employees, as well as moderating the relationship between social support and positive coping style. The current findings help to deepen the understanding of the relationship between social support and employment stress, and they have important implications for alleviating the employment stress of tourism employees in the context of the pandemic.

## 1. Introduction

The COVID-19 epidemic has now become one of the most widespread diseases in the world. As of 15 January 2023, over 662 million confirmed cases and over 6.7 million deaths have been reported globally [1]. Due to the contagious nature of the disease and the difficulty of controlling it. The economy under the impact of the epidemic has taken a major hit, leading to income uncertainty, employment difficulties, layoffs, and unemployment [2]. Mental health problems caused by employment stress are also paid attention to by the majority of scholars [3].

Tourism is one of the industries hardest hit by COVID-19 [4]. Due to the non-essential and vulnerable characteristics of tourism [5] coupled with the epidemic prevention and control that restricts people’s travel distance and prohibits mass gatherings [6], many tourism workers who make their living from tourism are facing unemployment and changing jobs, and the employment position provided by tourism has decreased from −164.506 million to −514.080 million [7]. The employment stress on tourism workers is relatively high in all industries. In addition to the mental pressure of reduced financial income and difficulty in maintaining family expenses, they have to upgrade their professional skills constantly and suffer from the pressure of possible elimination from the tourism industry. The mental health problems of this group of tourism workers need to be paid attention to by everyone.

Anxiety and depression brought about by employment stress affected people’s mental health and well-being, and social support helps buffer the negative effects of stressors [8]. The stress-buffering model [9] suggests that social support is beneficial in reducing social stress and promoting physical health in the face of external circumstances such as stress. In addition, internal individual factors such as psychological resilience and positive coping styles also influence employment stress. Social support, psychological resilience, and positive coping styles can all be effective in helping people recover from stressful situations such as illness and disaster [10]. In the field of psychology, previous studies have typically used the three as mediating variables to investigate their role in the prediction of psychological symptoms by social stress [11,12] or to study their relationship with happiness [3,13]. In contrast, these three psychological variables are less frequent in the field of tourism. In the context of the COVID-19 epidemic, the employment issues and mental health of college students have gained wider scholarly attention [14]. In contrast, less attention has been paid to the employment issues of tourism workers who were severely affected by the epidemic.

Therefore, this research takes tourism workers as the object for understanding the psychological state of tourism workers under the conditions of the epidemic through online and offline questionnaires and uses the structural equation model to explore the buffering effect of social support on tourism employment stress, as well as the role of psychological resilience and positive coping style in this process. It is hoped that this study can bring enlightenment to relieve the employment pressure on tourism workers.

## 2. Literature Review and Research Hypotheses

### 2.1. Social Support

Social support includes providing material resources and mental strength to cope with the epidemic [8], and support from family, friends and other external groups improves people’s stress resistance [15]. The concept of social support is often positive because it includes resources, encouragement, and recognition [16], research on social support involves “management”, “psychology”, “multidisciplinary psychology”, “applied psychology”, “biological psychology”, “clinical psychology” “experimental psychology”, “social psychology”, “nursing”, and “sociology”. In the field of nursing, the research focuses on the role of social support in the cure of disease [17]. In the field of sociology, the elderly, homosexuals, and other groups are the main research objects of social support. In particular, the social support role of social networks and social media has attracted more and more scholars’ attention [18]. In the field of psychology, the relationship between social support and depression [19], post-traumatic stress disorder [20], and stress are research hotspots.

The centrality of social support research is not only reflected in organizational research and the relationship of practitioners to work outcomes, health maintenance, and disease etiology [21], but also in the buffering effect of social support on stress. The stress-buffering model [22] suggests that social support is beneficial in reducing social stress and promoting physical health in the face of external circumstances such as stress. The theory has also been widely used in empirical studies, such as Zhang. RW’s research of college students’ self-representation on Facebook to obtain social support, and the behavior effectively buffered college students’ stress and improved life satisfaction [22]. Tuan’s research [23] found that family support has a positive impact on the post-traumatic growth of tourism workers, and the mediating mechanism in the positive stress mindset.

Social support can buffer the harmful effects of stress, which makes adverse reactions less severe. However, some scholars believe that social support can make stress worse [24,25], and the observed effects of social support are not all positive. This raises concerns, and it creates opportunities for future theoretical and empirical advances.

### 2.2. Positive Coping Styles

Research on positive coping styles includes “Psychology”, “Multidisciplinary”, “Nursing”, “Management”, “Education Educational Research”, “Health Care Sciences Services”, and “Public Environmental” Occupational Health” and other fields, Lazarus, R. S. divided coping styles into positive coping styles and passive coping styles [26], where positive coping means problem-solving strategies, and negative coping means emotion-centered coping mechanisms or non-coping [27]. Most of the findings suggest that positive coping styles can improve mental and physical health, while negative coping styles may lead to higher levels of psychological stress [28]

In the fields of nursing and medicine, the impact of positive coping styles on job satisfaction [29] and the role of positive coping styles in improving mental and physical health, preventing and alleviating negative psychological problems such as loneliness, depression, and stress [30] In the field of psychiatry, medical staff, the general population, and adolescents are the key research objects, and positive coping methods are considered to play a positive role in alleviating the anxiety of medical staff, maintaining mental health, and alleviating work pressure, and positive coping styles are one of the protective factors of adolescent mental health. In the fields of social psychology and management, the relationship between positive coping style and quality of life, emotional intelligence, self-acceptance, psychological toughness, social support, perceived stress, and happiness has become a hot research topic. In summary, positive coping is related to people’s psychosocial factors, mental health, work pressure, and job satisfaction. Zhu W et al. found that during the COVID-19 pandemic, medical staff in China felt highly anxious, positive coping styles were thought to play a mediating role in the relationship between social support and anxiety, and adequate social support and positive coping skills training could reduce anxiety among medical staff [31]. The study by Michelle et al. also found that positive coping styles (PCS) mediate the relationship between a positive psychological state (PsyCap) and employment stress and that PCS can strengthen the negative correlation between PsyCap and stress [32]. Gurvich et al. point out that positive emotion-focused coping strategies are effective in reducing psychological symptoms during the COVID-19 pandemic [33]. It can be seen that positive coping styles play a mediating role between psychosocial factors and mental health, and play a positive role in alleviating psychological problems.

### 2.3. Psychological Resilience

Psychological resilience is the ability of the individual to recover from a negative psychological state to live a normal life when faced with external circumstances such as disease, stress, trauma, and disaster [34]. Since the outbreak of COVID-19, the mental health of various groups has received attention, such as the stress problems of healthcare workers, people’s fear of COVID-19, and the depression of patients caused by isolation measures [3]. Psychological resilience has a strong positive impact on mental health during a pandemic, and enhancing psychological resilience can effectively buffer the negative impact of environmental stress on mental health. Thimbriel al. [35] studied chronic diseases and disabled people and found that people with disabilities with high psychological resilience were more likely to recover when faced with COVID-19 stress and unemployment. The positive role of psychological resilience in illness and mental health is evident. Therefore, the relationship between psychological resilience and social support, well-being, and satisfaction is also another important aspect of the study; Xiaohua Liu [36] found that social support and psychological resilience facilitated the alleviation of sleep disorders and mediated the relationship between perceived stress and sleep disorders. Psychological resilience [37] and coping with perceived stress mediate the relationship between fear of COVID-19 and happiness and stress. Moreover, psychological resilience and coping with perceived stress mitigate the impact of COVID-19 fear on happiness and stress.

### 2.4. Theoretical Model and Research Hypothesis

The stress-buffering model [9] suggests that social support can buffer individuals from the negative effects of stress when they are under stress and that this buffering process is achieved through changes in individual cognition and behavior. When individuals receive social support, they will reassess the potential harm caused by stress, improve their coping skills, and provide problem-solving solutions to reduce the impact of stress. Based on the buffer model of social support, the present study uses social support as the antecedent variable, positive coping style as the mediating variable, and employment stress as the outcome variable. Based on the related literature, the theoretical hypothesis is proposed.

This stress-buffering model has also been widely used in empirical studies, such as Zhang’s research of college students’ self-representation via Facebook to obtain social support, and the behavior effectively buffered college students’ stress and improved life satisfaction [22]. Therefore, in the current study, we constructed a hypothesis regarding the predicted relationship between social support and employment stress to address the question of whether a favorable social support system can provide adequate protection for the mental health of tourism employees when they face employment stress, thus reducing the negative effects of employment stress. The first hypothesis was as follows:

**Hypothesis 1** **(H1):**
*Social support has a significant negative effect on employment stress.*


Zimet divides the social support scale into family, friends, and other significant roles [15]. In this study, we classified social support into three dimensions: family support, friend support, and social group support. Broadly speaking, social groups refer to profit-making business organizations and non-profit government and industry associations. As a crucial part of the social support system of tourism employees, family, friends, and social groups all have different degrees of buffering effects on their psychological well-being. According to the findings of Kiernan, social support from family and friends can reduce the risk of bullying in adolescents [38]. In the present study, we constructed the following hypothesis regarding different dimensions of social support:

**Hypothesis 2** **(H2):**
*Family support has a significant negative effect on employment stress.*


**Hypothesis 3** **(H3):**
*Friend support has a significant negative effect on employment stress.*


**Hypothesis 4** **(H4):**
*Social group support has a significant negative effect on employment stress.*


Positive coping reflects the individual’s positive response to a stressful event by changing their state and actively seeking different ways to solve the problem. Wenting Bao [39] reported that college students in a stressful environment were more inclined to adopt a positive coping style after receiving social support from family and school. Therefore, we hypothesized that when tourism employees are in a stressful environment, the more support they receive from family, friends, and social groups, the more inclined they are to adopt positive coping styles. We constructed the following hypothesis:

**Hypothesis 5** **(H5):**
*Social support has a significant positive effect on positive coping style.*


**Hypothesis 6** **(H6):**
*Family support has a significant positive effect on positive coping style.*


**Hypothesis 7** **(H7):**
*Friends support has a significant positive effect on positive coping style.*


**Hypothesis 8** **(H8):**
*Social group support has a significant positive effect on positive coping style.*


Conventional positive coping styles include “talking to someone about your troubles” and “seeking advice from friends, relatives or classmates” to “solve problems” and relieve negative emotions such as stress [12], instead of avoiding the dilemma in other ways. Positive coping styles are associated with higher levels of positive cognitions in the face of stressful events [40]. We predicted that a relationship exists between positive coping styles and employment stress, constructing the following hypothesis:

**Hypothesis 9** **(H9):**
*Positive coping style has a significant negative impact on employment stress.*


According to the stress buffering model, the buffering effect of social support on stress is achieved through changes in individual cognition and behavior. According to this model, we assume that positive coping styles play a mediating role in social support. A study by Guo Xiaoli et al. found [41], positive coping style plays a complete mediating role between social support and mental health, and social support can improve the mental health of pediatric nurses by improving positive coping style. On the basis of empirical studies of positive coping styles, in the current study we hypothesized that positive coping styles play a mediating role in the relationship between social support and employment stress, as well as in the relationship between different dimensions of social support and employment stress. We constructed the following hypothesis:

**Hypothesis 10** **(H10):**
*Positive coping style mediates the relationship between social support and employment stress.*


**Hypothesis 11** **(H11):**
*Positive coping style mediates the relationship between family support and employment stress.*


**Hypothesis 12** **(H12):**
*Positive coping style mediates the relationship between friend support and employment stress.*


**Hypothesis 13** **(H13):**
*Positive coping style mediates the relationship between social group support and employment stress.*


The protection factor-protection factor model [42] says that there may be interactions between different protection factors, with the predictive effect of one protection factor on the outcome variable varying with differences in the level of another protection factor. Protective-protective factor model includes the exclusion hypothesis and promotion hypothesis. Psychological resilience, positive coping style, and social support are protective factors, and employment stress is the outcome variable. The enhancement or weakening of psychological resilience affects the predictive effects of social support on positive coping style, social support on employment stress, and positive coping style on employment stress. Based on the previous hypothesis and model, this study uses psychological resilience as a moderating variable.

According to the exclusion hypothesis in the protective factor-protective factor model [42], one protective factor may weaken the predictive effect of another protective factor on outcome variables, such as the effect of social support on loneliness is diminished when the psychological resilience of left-behind children is high [43]. Liu Huiying [44] found that in the relationship between cyberbullying incidents and psychological symptoms among college students, psychological resilience moderates the predictive effect of cyberbullying on psychological conditions, and psychological resilience may reduce the effect of cyberbullying on psychological symptoms. Based on the exclusion hypothesis in the protective factor-protective factor model and related studies, we speculate that the prediction of social support and its different dimensions on employment stress will decrease with the increase of psychological resilience, and make the following hypothesis:

**Hypothesis 14** **(H14):**
*Psychological resilience moderates the negative relationship between social support and employment stress.*


**Hypothesis 15** **(H15):**
*Psychological resilience moderates the negative relationship between family support and employment stress.*


**Hypothesis 16** **(H16):**
*Psychological resilience moderates the negative relationship between friend support and employment stress.*


**Hypothesis 17** **(H17):**
*Psychological resilience moderates the negative relationship between social group support and employment stress.*


According to the promotion hypothesis in the protective factor-protective factor model, one protective factor may enhance the predictive effect of another protective factor on the outcome variable [15]. For example, Meng Lin et al. [45] found that the emotional intelligence of college students with high resilience levels has a more significant positive predictive effect on social responsibility, that is, adding resilience as a moderating variable to the direct path between emotional intelligence and social responsibility enhances the overall predictive effect. Based on the positive nature of social support and positive coping styles, we speculate that the increase in psychological resilience may enhance the predictive effect of social support on positive coping styles, and also enhance the positive predictive relationship between family support, friend support, and social group support on positive coping styles. Specific hypotheses are as follows:

**Hypothesis 18** **(H18):**
*Psychological resilience moderates the positive relationship between social support and positive coping styles.*


**Hypothesis 19** **(H19):**
*Psychological resilience moderates the positive relationship between family support and positive coping styles.*


**Hypothesis 20** **(H20):**
*Psychological resilience moderates the positive relationship between friend support and positive coping styles.*


**Hypothesis 21** **(H21):**
*Psychological resilience moderates the positive relationship between social group support and positive coping styles.*


Liu, Xiaohua et al. [36] suggested that psychological resilience moderated the negative effects of perceived stress on sleep disturbance and that higher psychological resilience informed a smaller effect of stress on sleep disturbance. Based on the exclusion hypothesis of the protective factor, we hypothesized that the higher the psychological resilience, the smaller the effect of positive coping style on employment stress, and the lower the psychological resilience, the stronger the predictive effect of positive coping style on employment stress.

**Hypothesis 22** **(H22):**
*Psychological resilience moderates the negative relationship between positive coping styles and employment stress.*


On the basis of the above hypotheses, our research model appears in Figure 1.

## 3. Methods

### 3.1. Participants

This research focused on tourism employees as a target population, defined as individuals who engage in labor relations with tour operators and provide tourism services to tourists [46]. In the current study, past literature was synthesized with online information, and tourism employees were classified using two dimensions. One dimension was market operation subjects, which can be divided into the staff of tourist attractions, tourism hotels and lodges, travel agencies, tourism associations, tourism catering, and tourism enterprises. The second dimension was position categories, which can be divided into marketing, sales, management, accounting, and technical staff. The scope of the research object was limited according to these two classifications, and this was used to set the demographic variables of the questionnaire.

### 3.2. Measurements

#### 3.2.1. Demographic Variables

The demographic variables in this study mainly included age, gender, education level, job category, years of experience, and monthly income level. By counting the individuals of the participants, a certain degree of structural homogeneity of the sample is relatively ensured and over-concentration in one type of segmentation group is prevented. Of the participants, 145 were male (47.1%), and 163 were female (52.9%). In terms of education, 17.2%, 30.5%, 38.6%, 10.7%, and 2.9% of respondents had completed high school education or below, technical school education, bachelor’s degree, master’s degree, and doctorate degree, respectively, which is basically consistent with the actual situation of the education distribution of tourism employees. The respondents were aged 18~25 (16.2%), 26~30 (32.1%), 31~40 (23.4%), 41~50 (21.8%), and 51 years of age or older (6.5%). In terms of income, statistics show that 37% of the respondents have a monthly income of 5000 to 7000 yuan and 21.1% have a monthly income of 7000 to 10,000 yuan, and less than 5000 yuan and excess 10,000 yuan accounted for 27.3% and 14.6% respectively. Most respondents have been working for 6~10 years (52%), followed by those who have been working for 2~5 years (18.8%) and more than 10 years (18.2), and those who have been working for less than 1 year are 11%. In addition, respondents’ job categories include management (25.6%), operations (9.1%), technology (14.6%), marketing(29.5%), sales (15.3%), and others (5.8%).

#### 3.2.2. Social Support

On the basis of the “Social Support Scale” developed by Xiao Shuiyuan [47] in 1986 and a questionnaire designed by Li Liming [48], we designed an original questionnaire regarding the social support received by tourism employees in the present study, with three dimensions: family support (4 items), friend support (3 items), and social group support (3 items). The 10-item questionnaire was scored on a five-point Likert scale, with scores ranging from 1 to 5 on a scale from “totally inconsistent” to “complete conformity”. Higher scores indicated that the person received more social support. The Cronbach’s alpha value of the questionnaire was 0.9, which was consistent with the measurement requirements.

#### 3.2.3. Employment Stress

Although previous studies have examined employment stress among university student groups [49], there is relatively little research on tourism employees, and there are currently no scales for examining employment stress in this group. Referring to the idea of developing an employment stress questionnaire proposed in a master’s thesis by Chen Yuhong [50], in the present study we considered the specific situation regarding stress experienced by tourism employees and explored the sources of employment stress in the tourism industry during the pandemic, leading to the development of an employment stress questionnaire for tourism employees. The questionnaire included three dimensions: stress regarding employment situation and competition (3 items), stress regarding career quality evaluation (4 items), and stress regarding personal evaluation and family expectations (3 items). The scale consists of 10 items in total and adopts a five-point Likert scale, ranging from 1 to 5 on a scale of “totally inconsistent” to “complete conformity”. The Cronbach’s alpha value of the scale in this study was 0.911, indicating good reliability for use in stress measurement.

#### 3.2.4. Positive Coping Styles

On the basis of the Simple Coping Style Scale developed by Xie Yaning [51], in the current study, we used the dimension of positive coping styles to measure coping styles among tourism employees when faced with the stress of the pandemic with three measurement items. Participants answered these questions on a five-point Likert scale (1 = “totally inconsistent”, 5 = “complete conformity”), with higher total scores indicating that participants tended to adopt a positive coping style in the face of employment stress. The scale had good internal consistency (Cronbach’s alpha coefficient of 0.820), and good reliability.

#### 3.2.5. Psychological Resilience

In the current study, we adopted a simplified version of the psychological resilience scale (CD-RISC) translated and revised by Wang et al. [52]. This scale consists of nine items, including three dimensions of resilience (3 items), strength (3 items), and optimism (3 items). The scale is scored on a five-point Likert scale, ranging from 1 to 5 on a scale from “totally inconsistent” to “complete conformity”, with higher scores indicating higher levels of psychological resilience. The Cronbach’s alpha value of the scale in this study was 0.916, indicating a high level of reliability and suitability for use in this study.

### 3.3. Procedure

As shown in Table 1, we designed a preliminary questionnaire on the basis of realistic background, literature theory, and relevant mature scales. The first questionnaire was distributed offline to the staff of some tourism enterprises in Shenzhen from March 15 to March 18, 2022, and the results of the questionnaire were analyzed using reliability and principal component analysis, according to which the questionnaire items were adjusted and modified to form the final questionnaire. In order to make the origin of the research objects not limited to Shenzhen. The questionnaires were then distributed through social media platforms (Weibo, Zhihu, Xiaohongshu, and Momo) from 27 March to 2 April 2022, and 308 valid questionnaires were collected after eliminating invalid questionnaires, and the results were processed and analyzed. SPSS 23.0 was used for correlation analysis reliability testing and exploratory factor analysis Mplus 7.4 was used for confirmatory factor analysis, structural equation modeling, and the deviation corrected percentile bootstrap method for the mediating effect test, latent moderating effect test, and moderating/mediating effect test.

## 4. Results and Data Analysis

### 4.1. Correlation Analysis

The variables used in this study were continuous numerical variables, and Pearson’s correlation coefficients were used to analyze the correlation between social support, psychological resilience, positive coping style, and employment stress. The results revealed that social support was positively correlated with positive coping styles, and social support was negatively correlated with employment stress. A positive coping style was not directly correlated with psychological resilience. There was no significant correlation between positive coping style and psychological resilience but a negative correlation between positive coping style and employment stress. There is no significant correlation between psychological resilience and employment stress.

### 4.2. Reliability and Validity Analysis of the Questionnaire

Reliability analysis is an important method for testing the consistency or stability of survey results. We used Cronbach’s α coefficient as a measure of reliability. In this analysis, a larger reliability coefficient indicated greater credibility. A reliability coefficient value of 0.8 or higher was considered to be ideal, and a value of 0.7–0.8 was considered to be an acceptable range. In the questionnaire developed in this study, the total scale contained 32 items, and the value of Cronbach‘s α for the total scale was 0.803, indicating high reliability. The Cronbach’s α values for the social support subscale, positive coping style, psychological resilience, and the employment stress subscale were 0.900, 0.820, 0.916, and 0.911, respectively, which all met the criterion for acceptability (Cronbach’s α value greater than 0.7). In addition, the reliability of all 12 latent variables in the questionnaire was higher than 0.8, indicating that the reliability of the questionnaire was adequate and that the results of the measurement were credible.

Validity analysis was used to check whether the designed questions were reasonable and whether the set questionnaire achieved the purpose of the survey. In the current study, KMO and Bartlett’s in SPSS were used to check the suitability of the questionnaire for factor analysis, and exploratory factor analysis and confirmatory factor analysis were used to test the convergent validity and discriminant validity. The KMO statistic is an indicator for testing the bias correlation between variables. The closer the KMO value is to 1, the stronger the correlation between variables and the more suitable the original variables are for factor analysis. The KMO value in this study was 0.868, which met the criterion of a KMO value greater than 0.8.

Bartlett’s test of sphericity was used to test whether the correlation coefficient matrix was a unit matrix and whether the statistics obeyed the χ^2^ distribution, the null hypothesis was rejected when the probability of significance of the χ^2^ statistic value of Bartlett’s test of sphericity was *p* < 0.05, indicating that the correlation coefficient matrix could not be a unit matrix and factor analysis was appropriate. The results of the validity analysis of the total scale in this study revealed that KMO > 0.8 and *p* < 0.05, indicating that the total scale was suitable for factor analysis.

### 4.3. Factor Analysis

Because some of the questions in the scale were designed solely on the basis of the study content, it has not yet been confirmed that there is a favorable degree of interpretation and correspondence for the factors. Exploratory factor analysis conducted using principal component analysis and the varimax rotation technique indicated that 83.681% of the total variance was explained by 10 scale items.

On the basis of the exploratory factor analysis results, there was good correspondence between factors and measures, and the structural validity (including convergent validity and discriminant validity) and combined reliability of the scales was further tested using confirmatory factor analysis in Mplus7.4. As shown in Table 1, the factor loadings of all 32 measures were greater than 0.6, fulfilling the criteria for factor loadings. Thus, deletion of the measures was not required. The results of average variance extracted (AVE) can indicate convergent validity, emphasizing the measures that should be under the same factor. Table 2 shows that the AVE for all factors met the minimum criterion of 0.70, indicating that the convergent validity of each factor in this study was good. The composite reliability (CR) reflects whether all the measures in each factor consistently explained the factor, and the CR values for all 10 factors met the minimum criterion of 0.70, indicating that the combined reliability of all the factors in this study was good. In this study, the Fornell-Larcker criterion was used to test the discriminant validity of the scale, in which the square root of AVE captured the degree of variation in other variables explained by the latent variable. When the correlation coefficients between the latent variables are all smaller than the square root of AVE, the discriminant validity is ideal. The correlation coefficients of the latent variables are smaller than 0.5 and smaller than the square root of the corresponding AVE, indicating that the latent variables are correlated with each other and exhibit a certain degree of discrimination among them. Overall, the discriminant validity of the scale was satisfactory.

### 4.4. Structural Equation Model

In this study, structural equation model (SEM) analysis was conducted using Mplus7.4. The model fit was judged using five main indicators: χ^2^/df, comparative fit index (CFI), Tucker-Lewis index (TLI), root mean square of error of approximation (RMSEA), and standardized root mean squared residual (SRMR). The results of this study were as follows: χ^2^/df < 3, CFI > 0.90, TLI > 0.90, RMSEA < 0.05, and SRMR < 0.05, indicating that the initial model exhibited an ideal fit index.

### 4.5. Hypothesis Testing

The hypothesis relationship was tested by path analysis using the structural equation model. The hypothesis proposed that social support has a significantly negative effect on employment stress. Consistent with Hypothesis 1, social support was negatively related to employment stress (β = −0.571, *p* = 0.000). Hypothesis 2 predicted a negative relationship between family social support and employment stress. However, their interaction was not significant (β = −0.064, *p* = 0.393). Thus, Hypothesis 2 was not supported. Hypothesis 3 and 4 proposed that friend support and social group support had a significant negative effect on employment stress. Consistent with Hypotheses 3 and 4, friend support (β = −0.224, *p* = 0.002) and social group support (β = −0.317, *p* = 0.000) were negatively related to employment stress. As predicted by Hypothesis 5, social support was positively related to positive coping style (β = 0.464, *p* = 0.000). Family support, friend support, and social group support were included in social support, and Hypotheses 6, 7, and 8 proposed that all of these support types would have positive effects on positive coping styles. We found that family support was positively related to positive coping style (β = 0.204, *p* = 0.005), and friend support was positively related to positive coping style (β = 0.194, *p* = 0.008). Social group support was positively related to positive coping style (β = 0.109, *p* = 0.155). Consequently, Hypotheses 6 and 7 were supported, whereas Hypothesis 8 was rejected. Hypothesis 9 predicted a negative relationship between positive coping style and employment stress (β = −0.528, *p* = 0.000). Thus, Hypothesis 9 was supported.

#### 4.5.1. Testing for the Mediating Role of Positive Coping Style

We adopted the bootstrap method to test the significance of mediating paths, reflecting on whether the indirect effect of positive coping styles through social support on employment stress was significant. Path analysis was conducted using Mplus 7.4. The results are reported in Table 3.

As shown in Table 3 and Table 4, family support and employment stress were not directly related (*p* = 0.393), the indirect effect of family support on employment stress through positive coping style was significant (β = −0.158; *p* < 0.001), and the 95% confidence interval (CI) was (−0.238, −0.078), indicating that positive coping style had a complete mediating effect between family support and employment stress because the CI contained 0. The indirect effect of social support on employment stress through positive coping styles was significant (β = −0.154; *p* < 0.05), and the 95% CI was (−0.221, −0.064). The results shown in Table 5 indicate that Hypothesis 10, Hypothesis 11, Hypothesis 12, and Hypothesis 13 were supported. Furthermore, the sign of the path coefficients shows that the positive coping style plays a consistent role in all four paths, which enhances the total effect between the independent and dependent variables.

#### 4.5.2. Testing for the Moderating Role of Psychological Resilience

According to our research hypothesis and on the basis of the results of the reliability test indicating good reliability, we used the latent moderated structural equation method (LMS) to perform latent moderated model tests and mediator model tests with conditioning in structural equation models. The specific execution steps were as follows. Because the results obtained by the LMS method cannot output specific indicators of model fit, it was necessary to compare the benchmark model without interaction terms with the latent moderated model with interaction terms or the mediator model with moderated latent variables. The following expressions were simplified into the benchmark SEM model, the latent moderated model, and the moderated mediated SEM models. If the latent moderated model or the moderated mediated SEM model has a lower AIC value than the benchmark SEM model and a higher log-likelihood (H0) value than the benchmark SEM model, the latent moderated model or the moderated mediated SEM model has a better fit than the benchmark model.

Because the correlation between family support and employment stress and the correlation between social group support and positive coping style were not significant, the moderating effect of psychological resilience in these two paths was not discussed further, and Hypotheses 15 and 21 were not supported. The relationship between psychological resilience moderated social support and employment stress. First, the benchmark model without interaction terms was tested and the following metrics were obtained: χ^2^ = 119.638, df = 50, CFI = 0.937, TLI = 0.917, RMSEA = 0.083, SRMR = 0.078, χ^2^/df = 2.39, This indicates that the fitting degree of the benchmark model was at an acceptable level. The latent moderated model with interaction terms was then tested, and the AIC value of the latent moderated model was lower than that of the benchmark model, the log-likelihood value of the latent moderated model was higher than that of the benchmark model, and the degree of fit of the latent moderated model was reasonable.

As shown in Table 6 and Figure 2, the regression coefficient for the main effect was −0.619 (*p* < 0.001), and the coefficient of the interaction effect between psychological resilience and social support on employment stress was significant (β = 0.282, *p* < 0.001). This result confirmed the moderating effect of psychological resilience on the relationship between social support and employment stress, such that the higher the psychological resilience, the less the negative effect of social support on employment stress supporting Hypothesis 14. Following the same procedure, we tested the moderating effects of psychological resilience on the relationship between friend/social group support and employment stress. As shown in Table 6, the regression coefficient for the main effect was −0.386 (*p* < 0.01), and the regression coefficient for the interaction effect between friend support and psychological resilience on employment stress (β = 0.181, *p* < 0.01) was significant. Thus, H16 was supported. Furthermore, the regression coefficient for the main effect was −0.415 (*p* < 0.01), and the regression coefficient for the interaction effect between social group support and psychological resilience on employment stress was significant (β = 0.345, *p* < 0.001). Therefore, Hypothesis 17 was also supported.

The data shown in Table 6 are in accord with Hypothesis 18 to Hypothesis 21, indicating that when psychological resilience moderated the relationship between the mediator variable and the independent variable, the confidence interval, which excluded contain 0, was (0.161, 0.408). These findings indicated that there was a moderated mediation model. The main effect coefficient was 0.608, and the interaction coefficient was 0.285, *p* < 0.001, indicating that psychological resilience positively moderated the positive effect of social support on positive coping styles. Moreover, the mediation effect was enhanced, supporting Hypothesis 18. In the path of the relationship between family support and positive coping style moderated by psychological resilience, the confidence interval was (0.098, 0.332), which excluded 0; that is, there was a moderated mediation model. The main effect coefficient was 0.492, and the interaction coefficient was 0.215, *p* < 0.001, indicating that psychological resilience positively moderated the positive effect of family support on positive coping style. Moreover, the mediation effect was enhanced, supporting Hypothesis 19. However, the moderating effect of psychological resilience between friend support and positive coping style was not significant because the confidence interval contained 0 and *p* < 0.05; therefore, Hypothesis 20 was not supported. When psychological resilience moderated the relationship between the mediator variable and the outcome variable, the confidence interval contained 0 and *p* < 0.05, indicating that the moderating effect of psychological resilience in this path was not significant. Thus, Hypothesis 22 was not supported. The results of the above model fits are shown in Table 7.

In summary, psychological resilience plays a negative moderating role in social support and employment stress, friend support and employment stress, and social groups and employment stress. As shown in Figure 3, the higher the psychological resilience, the lower the predictive effect of social support on employment stress. Psychological resilience played a positive moderating role in social support and positive coping styles, and family support and positive coping styles, as shown in Figure 4, when the higher the psychological resilience, the stronger the predictive role of social support on positive coping styles.

## 5. Discussion

This study investigated the mediating effect of positive coping and the moderating effect of psychological resilience under the buffering effect of social support on employment stress during the COVID-19 pandemic. We constructed a moderating mediation model and tested 22 research hypotheses. Of these hypotheses, Hypothesis 16 was supported and Hypothesis 6 was not supported. Hypothesis 1, Hypotheses 3–7, Hypotheses 9–14, Hypotheses 16–19 were supported. Hypothesis 2, Hypothesis 8, Hypothesis 15, and Hypotheses 20–22 were not supported.

The establishment of Hypothesis 1, Hypothesis 3, and Hypothesis 4 prove the buffering effect of social support on stress, which is consistent with previous research results, indicating that the employment stress of tourism workers can be alleviated by enhancing social support. The establishment of H5–H7 shows that social support has a positive effect on positive coping style, the establishment of H9 shows that positive coping style has a significant negative effect on employment stress, and the establishment of H10–H13 shows that positive coping style plays an intermediary role in social support and employment stress. The establishment of the above hypothesis fully verifies the positive role of active coping, which is consistent with the conclusions of previous studies. It also suggests that tourism workers should adopt a positive coping style when facing employment stress. H14 and H16–H19 are supported, that is, psychological resilience mediates the positive relationship between social support and positive coping style, and the negative relationship between social support and employment stress. Consistent with previous research, the protective effect of resilience and its positive role in mental health have been reinforced.

In the following section, next, we will analyze and discuss the reasons or consequences of other hypotheses that were not supported.

Hypothesis 15 predicted that psychological resilience does not moderate the relationship between family support and employment stress, and it is not supported by the fact that Hypothesis 2 is not supported. That is, family support cannot directly buffer and reduce employee stress, possibly because tourism is a service industry, in addition to the high quality of middle and senior managers, most of the tourism employees such as tour guides, hotel service employees, and other front-line employees are not traditionally considered to have high social statuses in China. Thus, tourism employees tend to have a lower level of education and lower family economic status, as shown in the demographic characteristics of the sample in the current study. Therefore, even though China has a large population, there is still a shortage of labor force for some positions. For this segment of the travel industry [23], family support may have been primarily verbal encouragement and moral support rather than assistance with practical issues such as financial support, which may have been less effective for directly alleviating the employment stress of tourism employees. This also confirms the study of Tuan Trong Luu et al., that family support has a positive impact on the post-traumatic psychological growth of tourism workers, which reflects that even though Chinese family support cannot provide substantial employment stress for tourism workers, it is still beneficial to their mental health [23].

Hypothesis 21 predicted that psychological resilience does not moderate the relationship between social group support and positive coping, possibly because social group support does not directly contribute to positive coping (Hypothesis 8 was not supported). As seen in the scale question items, social group support includes substantial support such as government help policies, vocational skills training, and industry assistance. Although these support modalities should have facilitated positive coping among tourism employees, this was not the case. The low mean scores of the question items for social group support, as seen in the mean scale scores of the questionnaire, reflect the lack of support from social groups for tourism employees in China during the impacts of the COVID-19 epidemic. These findings indicate that there was no clarity regarding relevant support policies and industry association relief measures and that there is a need to raise awareness among Chinese tourism employees to seek support from social groups and optimize information channels.

Nevertheless, the important role that social support, positive coping, and psychological resilience play in managing employment stress can be seen in the current results. Social support, friend support, and social group support exhibited a direct buffering effect on employment stress, indicating that the buffering theory of social support is still applicable among tourism employees. Although family support does not directly buffer employment stress, it can facilitate positive coping by tourism employees, which can in turn alleviate employment stress. The mediating role of positive coping styles is also present in the relationships between social support, friend support, social group support, and employment stress. In addition, consistent with previous studies, individuals with high levels of psychological resilience are more likely to adopt positive coping styles to face adverse stressful situations.

## 6. Conclusions

In the context of the COVID-19 pandemic, the current study constructed an influence mechanism model of the effects of social support on employment stress among tourism employees on the basis of the buffering theory of social support, the protective factor-protective factor model, and the cognitive-interaction theory of stress. The results indicated that (1) social support, friend support and social group support have a buffering effect on employment stress; (2) positive coping plays a mediating role in the relationship between social support and employment stress, and positive coping plays a complete mediating role in the relationship between family support and employment stress, as well as a partially mediating role in the relationship between friend support, social group support, and employment stress; (3) the moderating role of psychological resilience weakens the negative influence of social support, friend support, and social group support on employment stress, with psychological resilience enhancing the positive influence of social support and family support on positive coping.

Overall, the results of this study largely support and validate the stress-buffering model. BLAIR WHEATON [54] argues that the concept of stress buffering is rich in connotations, and there are at least two distinct versions of stress buffering: two stress-buffering models and three other models. Two stress-buffering models were confirmed and supported, but three other models are sometimes thought to be coincident. Previous studies have mostly followed two stress-buffering models. Stress-buffering models are mostly used in the role of stress, illness or depression, and other psychological symptoms, and social support acts as a mediating or moderating variable. In contrast, this study explored and confirmed the buffering effect of social support on employment stress as an independent antecedent variable, in line with the three other models. In addition, family support did not have a buffering effect on employment stress, which also suggested that only certain sources of support had predictive power [55]. The application of the stress-buffering model in different contexts reveals that we need to be objective and accepting of the stress-buffering model, enriching the application scenarios. From a practical point of view, alleviating the employment stress of Chinese tourism workers requires some material social support, such as fiscal subsidies or social policy preferences. At the same time, spiritual strength should not be neglected. Tourism workers themselves need to maintain a positive and optimistic attitude, improve their psychological resilience, and adopt a positive way of coping with employment stress.

However, there are some shortcomings in this paper, the number of questionnaires is limited, the tourism employment situation in China has been in a dynamic development process, and there are some differences in the employment stress, coping styles, and social support of tourism workers in different periods, and the dynamic evolution mechanism of the buffering effect of social support on tourism employment stress can be explored in the future.

## Figures and Tables

**Figure 1 ijerph-20-02342-f001:**
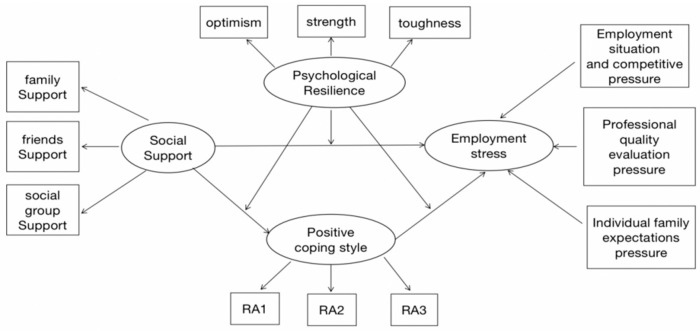
Conceptual model.

**Figure 2 ijerph-20-02342-f002:**
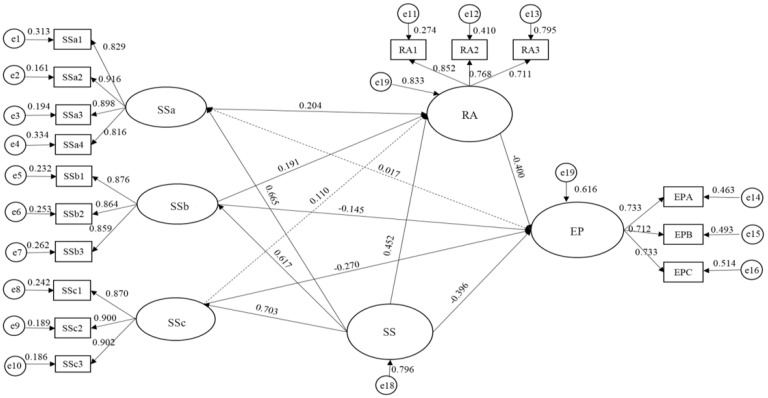
Path Results of Structural Equations.

**Figure 3 ijerph-20-02342-f003:**
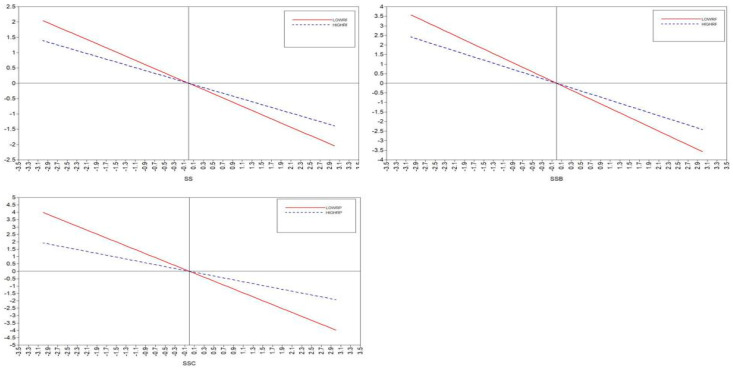
The moderating effect of psychological resilience on the relationship between social support and employment stress.

**Figure 4 ijerph-20-02342-f004:**
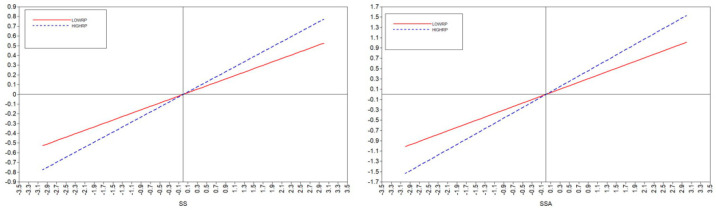
The moderating effect of psychological resilience on the relationship between social support and positive coping styles.

**Table 1 ijerph-20-02342-t001:** Variable measurement items.

Latent Variable	Items	Observed Variable	Reference
family support	SSa1	During the employment process, my family gave me financial support and other material help	XiaoShuiyuan [47]
	SSa2	In the process of employment, my family gave me spiritual encouragement.	
	SSa3	My family encouraged me to improve my employability and seize various opportunities to try.	Li Liming [50]
	SSa4	When I encountered difficulties in the employment process, I would confide in my family and seek help.	
friend support	SSb1	friends often help me relieve negative emotions in the employment process.	
	SSb2	In the process of employment, I will turn to my colleagues/friends for advice and help.	
	SSb3	In the process of employment, colleagues/friends provide me with industry information and employment guidance, and other practical help.	
Social group support	SSc1	Government agencies will implement corresponding supporting policies for employment in the tourism industry.	
	SSc2	Government/social groups provided employment counseling and vocational skills training during the pandemic.	
	SSc3	When encountering difficulties in employment, I will seek help from social organizations such as the Tourism Chamber of Commerce and employment Association.	
Positive Coping Styles	Ra1	I will continue to develop in the tourism industry	Xie Yaning [53]
	Ra2	I learned, developed, and improved my travel expertise during the pandemic.	
	Ra3	I tried various parergon to increase my income during the pandemic	
toughness	PR1	Even if there are obstacles, I believe I can achieve my goals.	Wang [23]
	PR2	I was able to think with concentration. in spite of stressful situations.	
	PR3	I can deal with some unpleasant or painful emotions, such as sadness, fear, anger, etc.	
strength	PR4	I was able to adapt to changes when they happened	
	PR5	After the trials, it will make me stronger.	
	PR6	After experiencing setbacks or hardships, I recover easily.	
optimism	PR7	Whatever comes my way, I can handle it.	
	PR8	When faced with problems, I try to see the humorous side of things.	
	PR9	At least I have people close to me who can help me in times of trouble.	
Employment situation and competitive stress	EPa1	I’m anxious about the current employment situation	ChenYuhong [52]
	EPa2	I don’t have the confidence to gain an edge in the rat race.	
	EPa3	I am concerned about the instability of the current tourism business.	
Professional quality evaluation stress	EPb1	Worried about my weak professional skills in the tourism industry	
	EPb2	I am worried that I will not be able to cope with the transformation and upgrading of the tourism industry	
	EPb3	I failed to obtain the relevant professional qualification certificate and lacked industry recognition.	
	EPb4	I think my own strength is not good enough, such as language expression ability and communication ability.	
Personal and family expectations stress	EPc1	I am worried about the economic benefits of the job and not being able to meet the expectations of my family.	
	EPc2	I worry about the lack of job security and a steady source of income.	
	EPc3	I worry about the low social recognition of my work and the mismatch in my expectations	

**Table 2 ijerph-20-02342-t002:** Confirmatory factor analysis parameter table.

Latent Variables	Observed Variables	Estimate	S.E	Est./S.E.	*p*-Value	AVE	CR
Family support	SSa1	0.828	0.021	39.888	***	0.7492	0.9226
	SSa2	0.916	0.013	70.043	***		
	SSa3	0.898	0.014	62.679	***		
	SSa4	0.816	0.022	37.174	***		
Friends support	SSb1	0.876	0.019	45.685	***	0.7506	0.9003
	SSb2	0.864	0.020	44.119	***		
	SSb3	0.859	0.020	42.488	***		
Social group support	SSc1	0.870	0.018	49.197	***	0.7935	0.9202
	SSc2	0.900	0.015	59.676	***		
	SSc3	0.902	0.015	60.438	***		
Toughness	PRa1	0.950	0.011	86.735	***	0.8136	0.9290
	PRa2	0.873	0.016	53.680	***		
	PRa3	0.881	0.016	56.108	***		
Strength	PRb1	0.924	0.013	68.652	***	0.7968	0.9216
	PRb2	0.886	0.016	55.827	***		
	PRb3	0.867	0.018	49.399	***		
Optimism	PRc1	0.857	0.019	45.400	***	0.7757	0.9120
	PRc2	0.917	0.015	62.131	***		
	PRc3	0.867	0.018	47.771	***		
Employment situation and competitive stress	EPa1	0.900	0.016	55.707	***	0.7683	0.9086
	EPa2	0.873	0.018	48.156	***		
	EPa3	0.856	0.019	44.465	***		
Professional quality evaluation stress	EPb1	0.775	0.026	30.297	***	0.7249	0.9131
	EPb2	0.866	0.018	48.449	***		
	EPb3	0.889	0.016	55.728	***		
	EPb4	0.871	0.017	50.442	***		
Personal and family expectation pressure	EPc1	0.912	0.015	60.569	***	0.7778	0.9130
	EPc2	0.866	0.018	47.376	***		
	EPc3	0.867	0.018	47.850	***		
Positive coping style	RA1	0.853	0.029	29.291	***	0.6067	0.8215
	RA2	0.768	0.033	23.576	***		
	RA3	0.709	0.036	19.733	***		

Note. CR = composite reliability, AVE = average variance extracted, *** means *p* < 0.001.

**Table 3 ijerph-20-02342-t003:** Normalized path coefficients and significance.

Path	Path Coefficient	Standard Error	T	*p*	Significance
SS→EP	−0.571	0.061	−9.386	0.000	significant
SSa→EP	−0.064	0.076	−0.854	0.393	non-significant
SSb→EP	−0.224	0.074	−3.039	0.002	significant
SSc→EP	−0.317	0.076	−4.158	0.000	significant
SS→RA	0.464	0.063	7.386	0.000	significant
SSa→RA	0.204	0.073	2.780	0.005	significant
SSb→RA	0.194	0.073	2.646	0.008	significant
SSc→RA	0.109	0.077	1.421	0.155	non-significant
RA→EP	−0.528	0.057	−9.323	0.000	significant

SS means social support; SSa means family support; SSb means friend support; SSc means social group support; EP means employment stress; RA means positive coping styles.

**Table 4 ijerph-20-02342-t004:** Mediating effect (Bootstrap = 1000).

Path	Direct Effect	Mediating Effect	Proportion	Standard Error	T	*p*	Boot 95%CI
SSa→RA→EP	−0.162	−0.158	0.494	0.041	−3.885	***	[−0.238,−0.078]
SSb→RA→EP	−0.233	−0.142	0.379	0.040	−3.556	***	[−0.221,−0.064]
SSc→RA→EP	−0.308	−0.127	0.292	0.034	−3.708	***	[−0.194,−0.060]
SS→RA→EP	−0.413	−0.154	0.272	0.039	−3.966	***	[−0.211,−0.069]

Note. *** means *p* < 0.001.

**Table 5 ijerph-20-02342-t005:** Comparison between the benchmark model and latent moderated model.

Moderated Path	Fit Index	Benchmark Model	Latent Moderated Model	Model Compare
SS→EP	AIC HO	16,243.929 −8081.964	16,231.188 −8074.594	better better
benchmark model fit index	χ^2^ = 119.638; df = 50; CFI = 0.937; TLI = 0.917; RMSEA = 0.083; SRMR = 0.078; χ^2^/df = 2.39; acceptable			
SSb→EP	AIC HO	13,767.411 −6843.705	13,703.235 −6809.617	better better
benchmark model fit index	χ^2^ = 143.935; df = 50; CFI = 0.937; TLI = 0.916; RMSEA = 0.093; SRMR = 0.098; χ^2^/df = 2.88; acceptable			
SSc→EP	AIC HO	13,852.738 −6886.369	13,774.530 −6845.265	better better
benchmark model	χ^2^ = 140.534; df = 50; CFI = 0.943; TLI = 0.925; RMSEA = 0.092;			
fit index	SRMR = 0.096; χ^2^/df = 2.81; acceptable			

**Table 6 ijerph-20-02342-t006:** Moderating effect.

Moderating Path	Main Effect Coefficient	Interaction Effect Coefficient	Estimation of Standardization	95% Confidence Interval
SS→EP	−0.619 ***	0.282 ***	0.391 ***	[0.111,0.407]
SSb→EP	−0.386 ***	0.181 **	0.201 **	[0.051,0.312]
SSc→EP	−0.415 **	0.345 ***	0.415 ***	[0.211,0.479]
SS→RA	0.608 ***	0.285***	0.144 ***	[0.161,0.408]
SSa→RA	0.492 ***	0.215***	0.094 **	[0.098,0.332]
SSb→RA	0.373 ***	0.041 ***	0.046	[−0.022,0.029]
RA→EP	−0.482 ***	0.124 ***	0.167	[−0.014,0.262]

Note. *** means *p* < 0.001, ** means *p* < 0.01.

**Table 7 ijerph-20-02342-t007:** Comparison between the benchmark model and moderated mediation model.

Moderated Path	Fit Index	Benchmark Model	Moderated Mediation Model	Model Compare
SS→RA	AIC HO	16,181.758 −8050.879	16,167.778 −8042.889	better better
benchmark model fit index	χ^2^ = 57.467; df = 50; CFI = 0.993; TLI = 0.991; RMSEA = 0.044; SRMR = 0.044; χ^2^/df = 1. 15; fine			
SSa→RA	AIC HO	14,323.004 −7118.502	14,313.492 −7112.746	better better
benchmark model fit index	χ^2^ = 70.747; df = 61; CFI = 0.995; TLI = 0.993; RMSEA = 0.043; SRMR = 0.040; χ^2^/df = 1.16; fine			
SSb→RA	AIC HO	13,674.763 −6797.382	13,674.298 −6796.149	better better
benchmark model fit index	χ^2^ = 51.287; df = 50; CFI = 0.999; TLI = 0.999; RMSEA = 0.038; SRMR = 0.039; χ^2^/df = 1.03; fine			
RA→EP	AIC HO	16,174.145 −8045.073	16,172.941 −8043.471	better better
benchmark model fit index	χ^2^ = 45.854; df = 48; CFI = 1; TLI = 1.003; RMSEA = 0.035;SRMR = 0.029; χ^2^/df = 0.96;fine			

## Data Availability

The datasets generated during and/or analyzed during the current study are available from the corresponding author on reasonable request. The specifific questionnaires (Chinese version) used in this study are available on request from the corresponding author.

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
