# Peer review of "Stress-Buffering Effects of Social Support on Tourism Employees during the COVID-19 Pandemic: A Moderated Mediation Model"

_ijerph, 2023, doi:10.3390/ijerph20032342_

Round 1

Reviewer 1 Report

This paper proposes an original study that deals with a very interesting topic. Although this article is original, this reviewer suggests a revision.

Section Introduction: the introduction should be revised. It needs a clean-up, is too long. The authors should be more clear and direct. At the moment, the front end of the paper is quite vague and one does not get a sense that this kind of study is really needed. A better way to begin this would be to talk about the problem in the first part (explain why this research is relevant), give a brief overview of past studies in the second, show what you are doing different in the third, and use the fourth for the main goals and novelty of the study.

The authors should consider including a Review Literature section with the most relevant studies on the topic, the hypothesis development and models. It is not correct to include this content in the methodology. Also, some of the hypothesis need a better ground and literature support, namely: H5-H8, H10-H13 and H18-H21.

In the methodology section, authors should include the following information more or less by this order: research setting, survey instrument, sampling, data collection procedure and sample profile.

Section Discussion: all the results should be discussed.

Section Conclusion: the results are very interesting hence the conclusions are extremely poor. The authors should include the most relevant implications for theory and practice. Also, the authors should add some limitations and further research.

Some typos and grammar errors should be corrected. Also, some relevant literature on the COVID-19 impact in tourism is not referenced.

Author Response

REVIEWER 1 COMMENT

AUTHOR RESPONSE

PAGE NUMBER

Section introduction:

the introduction needs a clean-up, is should be more clear and direct. At the moment, the front end of the paper is quite vague and one does not get a sense that this kind of study is really needed.

Thank you so much for your pertinent and valuable comments. We have revised the introduction section to make it more concise and direct to present the problem and the need to solve it.

The authors should consider including a Review Literature section with the most relevant studies on the topic, the hypothesis development and models. Also, some of the hypothesis need a better ground and literature support, namely: H5-H8, H10-H13 and H18-H21.

We appreciate your bringing to our attention the need to add a literature review section to discuss the literature on related topics. We believe that the addition of this section will help readers understand the connotation and function of professional terms such as social support, psychological resilience and positive coping style, and make the hypothesis section seem reasonable. It should be mentioned that the hypothesis model (such as protection factor-protection factor model) was proposed with reference to some authoritative Chinese literature. Due to the short time of this revision, the content of this part really needs to be improved further.

pp.2-4

In the methodology section, authors should include the following information more or less by this order: research setting, survey instrument, sampling, data collection procedure and sample profile.

We appreciate your helpful remarks.

We have improved and supplemented relevant contents in two parts: Procedure and Demographic Variables

pp.4、8

Section Discussion: all the results should be discussed

Thanks so much for your comment. We have made changes to improve.

pp.16

Section Conclusion: the conclusions  should include the most relevant implications for theory and practice. Also, the authors should add some limitations and further research.

Many thanks. We have made changes to improve.

pp. 17

Reviewer 2 Report

I am very honor to review the important and intresting study. Thank you very much. However, I have some concerns as follow.

Introduction

1. Please update the data.

2. The introduction section lacks much information. So I suggest rewriting the introduction section. The introduction may include five sub-sections. The first part proposes the research topic and introduces the importance of the research topic. The second part carries on the literature review of research topic and points out the research gaps. The third part puts forward some research questions aiming at the research gaps. The fourth part points out the contribution of this research after solving the research questions. The fifth part points out the layout of the article. However, it seems that the introduction did not clearly present the research gap. Meanwhile, why did you select positive coping style as mediating variable and psychological resilience as moderator?

Method

1. Is the second part called Method appropriate?

2. What is the basic theory of model building?

3. The proposing process of hypothesis is lack of sufficient literature evidence and theoretical support. For example, hypothesis 1 is based on a single reference.

Results and data analysis

1. The common method bais is missing.

2. It is suggested to present the correlations between variables.

Discussion

Discussion is a dialogue with the literature. But the fourth part of the discussion does not seem to have any references.

It is suggested to add the parts of theoratical implication, pratical implication and limitations and further research.

Author Response

The introduction section lacks much information.

Thank you so much for your pertinent and valuable comments. We have revised the introduction section and add a literature review section

Method: The proposing process of hypothesis is lack of sufficient literature evidence and theoretical support.

Many thanks.We add a literature review section after the introduction, hoping to enrich the theoretical background of this paper, the theoretical model is on the basis of the buffering theory of social support, the protective factor-protective factor model, and the cognitive-interaction theory of stress. Due to the short time of this revision, the content of this part really needs to be improved futher.

pp.4、8

Discussion is a dialogue with the literature. But the fourth part of the discussion does not seem to have any references

We appreciate your helpful remarks.We added those references.

Sunny Sun, Zhaoping (George) Liu, Rob Law & Shiyun (Eva) Zhong (2017) Exploring human resource challenges in China’s tourism industry, Tourism Recreation Research, 42:1, 72-83.

Tuan Trong Luu ,Family support and posttraumatic growth among tourism workers during the COVID-19 shutdown: The role of positive stress mindset, Tourism Management, 2022.

pp.16

Reviewer 3 Report

The article makes an adequate use of multivariate analysis, fully justifying its validity and reliability, as well as making very clear the relevance of the use of structural equations. But the main problem of the article is the quality of its data (your results are only as good as the quality of the data).

A sample of 308 questionnaires:

Is not representative of the collective of tourism workers in china. 

The questionnaire is not shown anywhere in the article.

A web survey on social networks is answered only by those people with more interest in the subject.

The software used for data collection is not indicated either.

I also think that the authors should review the criteria to be used in a web survey.

Finally, it should be pointed out that in the discussion the authors make some interpretations of the hypothesis that are not supported by previous results. From my point of view they should be based on something more and not just speculate.

Author Response

REVIEWER  3  COMMENT

AUTHOR RESPONSE

PAGE NUMBER

The date is not representative of the collective of tourism workers in china.

Many thanks.In terms of the number of questionnaires, 308 valid data is not very rich. It is not an easy thing to find more tourism workers as new survey objects to obtain new data in a short time. In order to make the data more representative and persuasive, we have made more complete supplements to data sources, sample screening criteria and analysis in Procedure and Demographic Variables

pp.4、8

The literature review is insufficient and confusing, since it is not well structured by technique, and it also does not discriminate the different contribution of the different studies.

We appreciate your helpful remarks.We have added a literature review section, and we look forward to your comments for further improvement

pp. 2-4

it should be pointed out that in the discussion the authors make some interpretations of the hypothesis that are not supported by previous results

We appreciate your comments. In the discussion and conclusion of literature, the discussion part is more reasonable rather than just based on speculation through dialogue with other literature.

pp.16

Round 2

Reviewer 1 Report

Authors made significant efforts to improve the paper. The paper presents an interesting study.

Author Response

Thank you so much for your pertinent and valuable comments.

Reviewer 2 Report

Although the authors have made systematic improvement to the paper, however, I still have some concerns:

1. Please give more theoritical and literatural evidence and justification when you propose the hypothesis

2. In the conclusion section, the theoritical and practical implications are necessary.

Author Response

REVIEWER 2 COMMENT

AUTHOR RESPONSE

PAGE NUMBER

Please give more theoritical and literatural evidence and justification when you propose the hypothesis

Thank you so much for your pertinent and valuable comments. We added corresponding theories and more literature to support our hypothesis. Specifically, the stress buffering model explains the hypothesis of the relationship between social support and employment stress and the mediating role of positive coping style, and the protective-protective factor theory explains the moderating role of psychological resilience, and adds more literature to make the theoretical model convincing. In addition, to make the article more logically coherent, we adjust the research hypothesis and theoretical model to the second section.

pp.4-6

In the conclusion section, the theoritical and practical implications are necessary.

We appreciate your helpful remarks. We have added the corresponding content in the conclusion section.

pp.21

Reviewer 3 Report

Dear authors:

The revisions made are correct, so I believe that the article can be published.

Regards.

Author Response

(The authors gave the same response as above.)
